# Ultraconserved bacteriophage genome sequence identified in 1300-year-old human palaeofaeces

Piotr Rozwalak[1], Jakub Barylski [2], Yasas Wijesekara[3], Bas E. Dutilh [4,5] & Andrzej Zielezinski [1]

Bacteriophages are widely recognised as rapidly evolving biological entities. However, knowledge about ancient bacteriophages is limited. Here, we analyse DNA sequence datasets previously generated from ancient palaeofaeces and human gut-content samples, and identify an ancient phage genome nearly identical to present-day *Mushuvirus mushu*, a virus that infects gut commensal bacteria. The DNA damage patterns of the genome are consistent with its ancient origin and, despite 1300 years of evolution, the ancient *Mushuvirus* genome shares 97.7% nucleotide identity with its modern counterpart, indicating a long-term relationship between the prophage and its host. In addition, we reconstruct and authenticate 297 other phage genomes from the last 5300 years, including those belonging to unknown families. Our findings demonstrate the feasibility of reconstructing ancient phage genome sequences, thus expanding the known virosphere and offering insights into phage-bacteria interactions spanning several millennia.

Bacteriophages diversified over billions of years through a co-evolutionary arms race with their microbial hosts[1]. Due to recent advancements in metagenomic sequencing[2] and computational analysis[3], the vast genomic diversity of phages can now be explored, catalogued[4], or even tracked through space[5] and time[6]. However, most studies only sample present-day phages, thus lacking an evolutionary perspective. The longest study of substitutions and recombinations in phage genomes spans nearly three decades[7], representing only a partial glimpse into the complex evolutionary history of bacteriophages. To fully understand the phylogeny of these viruses and their effect on microbial ecosystems, it is essential to go further back in time.

Previous research on ancient virology has mainly focused on the reference-based reconstruction of human pathogens[8], retroviral elements incorporated in animal genomes[9], or eukaryotic viruses that have remained dormant since prehistory[10]. However, knowledge about ancient bacteriophages is limited[11]. The first reports on ancient phages described viruses from 14th-century faecal material in Belgium[12] and the gut contents of pre-Columbian Andean mummies[13]. However, these studies lacked authentication based on terminal deamination patterns observed in ancient DNA (aDNA)[14], which is used to distinguish ancient sequences from modern contaminants[15]. Furthermore, the small amount of sequenced DNA in these investigations resulted in the recovery of only fragmented phage genomes. To our knowledge, only one complete oral phage genome has been published that meets the criteria of authenticated aDNA reconstruction[16].

While the recovery of hundreds of high-quality ancient bacterial genomes is now possible[17,18], their phage counterparts remain largely unexplored. However, recent developments in computational tools for viral metagenomics[3,4], combined with the extraction of exceptionally well-preserved ancient DNA samples[17,19–23] offer new opportunities to unlock the mystery of past phage genome diversity.

[1]Department of Computational Biology, Faculty of Biology, Adam Mickiewicz University, Poznan 61-614, Poland. [2]Department of Molecular Virology, Faculty of Biology, Adam Mickiewicz University, Poznan 61-614, Poland. [3]Institute of Bioinformatics, University Medicine Greifswald, Felix-Hausdorff-Str. 8, 17475 Greifswald, Germany. [4]Institute of Biodiversity, Faculty of Biological Sciences, Cluster of Excellence Balance of the Microverse, Friedrich Schiller University Jena, 07743 Jena, Germany. [5]Theoretical Biology and Bioinformatics, Science4Life, Utrecht University, Padualaan 8, 3584 CH Utrecht, the Netherlands. ✉e-mail: bedutilh@gmail.com; andrzej.zielezinski@amu.edu.pl

Here, we analyse draft and complete genomes assembled from 150–5300 years old palaeofaeces and human gut-content samples[17,19–23]. We use this collection to (i) identify 298 ancient phage genomes, (ii) authenticate their ancient origin based on DNA damage patterns, (iii) determine their taxonomic assignments and relationships to modern viruses, (iv) predict hosts using state-of-the-art bioinformatic tools, and finally (v) characterise the particularly stable genome of one encountered virus, *Mushuvirus mushu* that is 97.7% identical to its present-day reference. Together, these results demonstrate that the large-scale recovery of virus genomes from ancient samples is possible and may provide unexpected insights into the evolutionary history of the human virome.

## Results

### Identification and validation of ancient phage genomes

We selected aDNA sequence datasets from 30 samples previously published in studies on the ancestral human gut microbiome[17,19–23]. These samples were derived from eight sites in Europe and North America, dating back between 150 and 5300 years ago, using the C14 method (Supplementary Data 1). The de novo assembly of selected libraries resulted in 72,693 high-quality contigs (mean length: 12,592 nucleotides ± 59.3 standard error; Fig. 1a). However, we deemed less than half of the assembled contigs as ancient based on deamination patterns observed at the 5′ ends of the sequencing reads (see Methods). A total of 2375 sequences were classified as viral by at least two methods (VIBRANT[24], VirSorter2[25] and Jaeger; Supplementary Data 2) and were selected as bona fide virus contigs. Among these, 383 were at least 20 kb long or assessed by CheckV as either medium or better quality (>50% completeness; Fig. 1b and Supplementary Data 3) and selected for further analysis. We clustered the selected sequences into 298 species-level viral operational taxonomic units (hereafter referred to as aMGVs - ancient metagenomic gut viruses) based on 95% average nucleotide identity (ANI) with over 85% alignment fraction[26]. Despite fragmentation and degradation of the aDNA, our aMGV collection had size range from 5 kbp to 292 kbp, an estimated mean completeness of 50.3%, and comprised of 49 high-quality or complete genomes (Supplementary Data 3). Most of aMGVs (59%) are likely lytic, whereas the remaining 41% of viruses are potentially lysogenic (Supplementary Data 4). Although the mean damage at the first position of reads mapping to collected aMGVs was relatively low (0.025 frequency), it

was 12 times higher than the control modern viral genomes from the human gut (0.002 frequency; Fig. 1c, d and Supplementary Data 3).

### Human gut origin of ancient phage genomes

Deamination damage patterns is a well-established technique for ancient DNA validation, but the complexity of ancient metagenomic data can necessitate the implementation of additional authentication steps. In our case, we decided to rule out post-depositional contamination with DNA from the environment (e.g., cave sediments), which may also exhibit damage due to the same degradation processes as the aDNA of interest[27]. To estimate the risk of such contamination, we constructed a gene-sharing network using vContact2[28] to phylogenetically relate ancient and modern phages (Fig. 2a), including reference viruses from different environments in the IMG/VR database[29] and prokaryotic viruses classified by the International Committee of Virus Taxonomy (ICTV)[30]. In the network (Supplementary Data 5), 151 ancient phages were distributed across 122 viral clusters (VCs). These VCs were dominated by mammalian-gut-associated viruses from IMG/VR (80%, see Fig. 2b), with a high representation (68%) of phages found in humans. The estimated contamination was relatively low; only 5% of aMGVs (n = 15) clustered with viruses from environmental or engineered ecosystems. We recognised that 49% of aMGVs (n = 147) did not cluster with any modern viruses. The mean ANI of all aMGVs to the closest IMG/VR and GenBank viral genome was 40% (± 24.7%, Supplementary Data 6 and 7). Only one outlier aMGV, *Mushuvirus mushu*, had more than 95% ANI, and it is described in further sections (Supplementary Fig. 1).

Further evidence supporting the ancient gut origin of the 298 analysed aMGVs came from the distribution of their predicted hosts (see Methods; Supplementary Fig. 2 and Supplementary Data 8). Over half of the ancient viruses were predicted to infect *Clostridia* and *Bacteroidia* hosts − two dominant classes in the gut microbiome[31]. The distribution of host bacteria assigned to ancient viruses was more similar to that of modern viruses from the digestive systems of different animals than viruses of other environments. For example, we observed a strong correlation (Pearson's r = 0.95, P = 3.29 × 10⁻⁶) between the proportion of host classes of ancient phages and hosts of modern viruses in human large intestines (Supplementary Data 9). Hence, both the results of host prediction and the gene-sharing network analysis suggest that the ancient phage sequences primarily

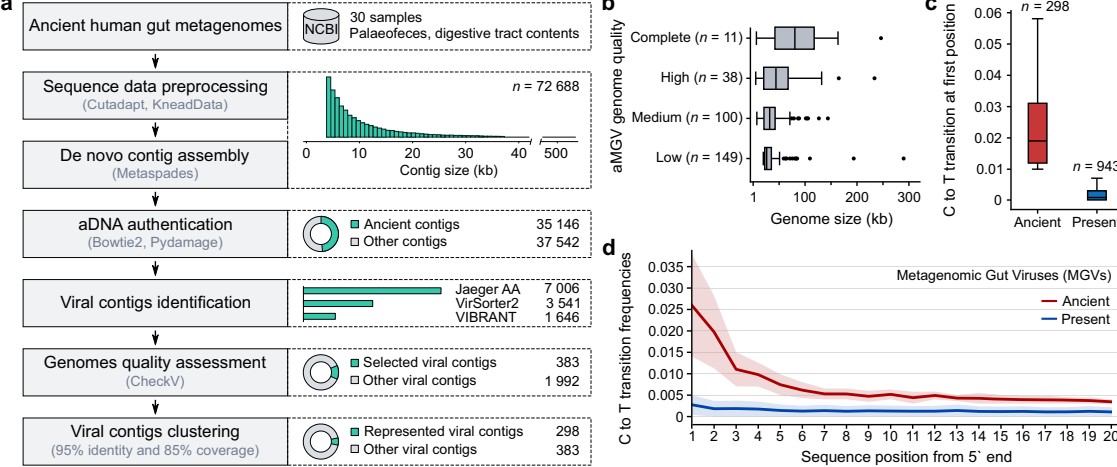

**Fig. 1 | Identification of phage genomes in ancient human gut metagenomes.** **a** Overview of workflow to identify ancient phage genomes. **b** Distribution of the genome sizes of 298 ancient metagenomic viruses (aMGVs) stratified by CheckV[61] genome quality. **c** Distribution of C → T substitution frequencies at the first position of the 5′ end of sequencing reads from ancient (n = 298, red) and present-day (n = 943, blue) phage contigs. Box and whisker plots in **c** and **d** show median (centre

line), upper and lower quartiles (represented by boxes), and highest (upper whisker) or lowest (lower whisker) value within a 1.5 inter-quartile range (IQR) while black dots indicate values outside of the IQR. **d** Comparison of damage patterns between selected modern viral genomes from MGV[31] (blue) and aMGVs (red). The solid line shows the mean frequency of C → T substitutions and the shade indicates the standard deviation. Source data are provided as a Source Data file.

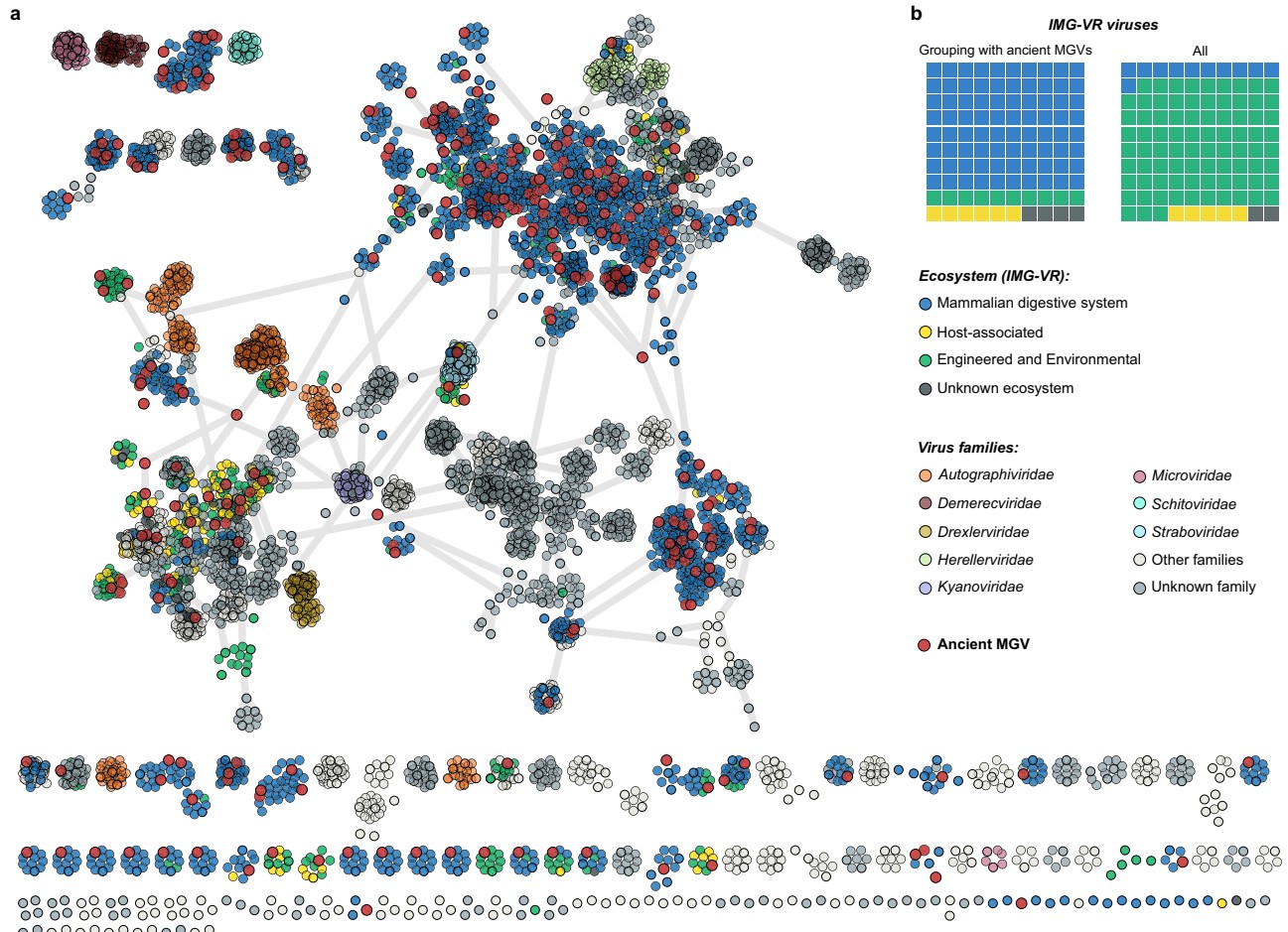

**Fig. 2 | Ancient phage genomes from palaeofaeces are related to currently known mammalian-gut-associated viruses. a** vContact2 gene-sharing network[28] of 151 ancient metagenomic gut virus (aMGV) genomes (red circles), 2198 selected close relatives from the IMG/VR database, and 3655 prokaryotic viruses classified by the International Committee of Virus Taxonomy. Distantly related aMGVs (*n* = 147) were outliers and not included in the gene-sharing network. **b** Waffle charts represent the proportion of contemporary viruses categorised by ecosystems in two data sets: clusters comprising aMGVs (left) and the entire IMG/VR database. Source data are provided as a Source Data file.

originated from the human gut rather than the surrounding environment.

## Catalogue of the human gut virome expands into the past

We performed geNomad's[32] marker-based taxonomic assignment for all 298 ancient viral sequences, 293 of which (~98%) were assigned to the *Caudoviricetes* class, three to the *Megaviricetes* class (nucleo-cytoplasmic large DNA viruses - NCLDV), and two single-strand viruses to the *Cressdnaviricota* and *Phixviricota* phyla (Supplementary Data 10). Most aMGVs from the *Caudoviricetes* class were not classified at lower taxonomic ranks, except for nine that were assigned to the order *Crassvirales* and the families *Autographiviridae*, *Straboviridae*, *Herelleviridae*, and *Rountreeviridae*. To improve the taxonomic resolution, we selected 49 high-quality or complete ancient genomes and clustered them with RefSeq and IMG/VR virus genomes based on pairwise average amino acid identity (AAI) and gene sharing, followed by manual assignment to the ICTV taxonomy (see "Methods" section). Additionally, we constructed a proteomic tree to support the clustering results and illustrate the relationships between the modern and ancient viruses (Fig. 3). This analysis revealed that high-quality or complete aMGVs were distributed across 39 putative families and 46 putative genera (Supplementary Data 11).

Most putative families (90%, 35 out of 39) and genera (72%, 33 out of 46) contained aMGVs and modern representatives from IMG/VR and/or RefSeq. Only a small proportion of those groups have been classified by ICTV, including six families (15%, *Straboviridae*, *Peduoviridae*, *Casjensviridae*, *Microviridae*, *Chaseviridae*, and *Guelinviridae*), and eight genera (17%, *Tequatrovirus*, *Plaisancevirus*, *Pbunavirus*, *Astrithrvirus*, *Brucesealvirus*, *Loughboroughvirus*, and *Mushuvirus*). Most of these putative taxa included a single ancient phage grouped with multiple contemporary viruses. An interesting example of such a grouping is a clade of *Escherichia*-infecting members of the *Tequatrovirus* genus (including bacteriophage T4, Supplementary Fig. 3a) with an ancient virus located on the ancestral branch. This ancient *Tequatrovirus* maintains genome organisation and conserved structural proteins, with ANI of 88.1% and AAI of 94.7% to its closest modern relative, *Tequatrovirus cromcrrp10* (Supplementary Fig. 3b). Only a few putative taxa contained multiple ancient representatives. For example, two genus-like groups ("13" and "14" see Supplementary Data 11) formed a family-like unit that encompassed three ancient phages, 36 IMG/VR viruses and the *Salmonella*-infecting phage, *Astrithrvirus astrithr*.

In contrast to groups with multiple representatives, there were four putative families and 13 genera represented by only a single aMGV. While these phages could hypothetically represent extinct lineages, it is likely that modern representatives of those groups have yet to be discovered. Although it is beyond the scope of the current study, one way to address this question could be to identify conserved marker regions in the ancient genomes and attempt to amplify them in locations where many modern human gut phages come together, such as sewage systems[5].

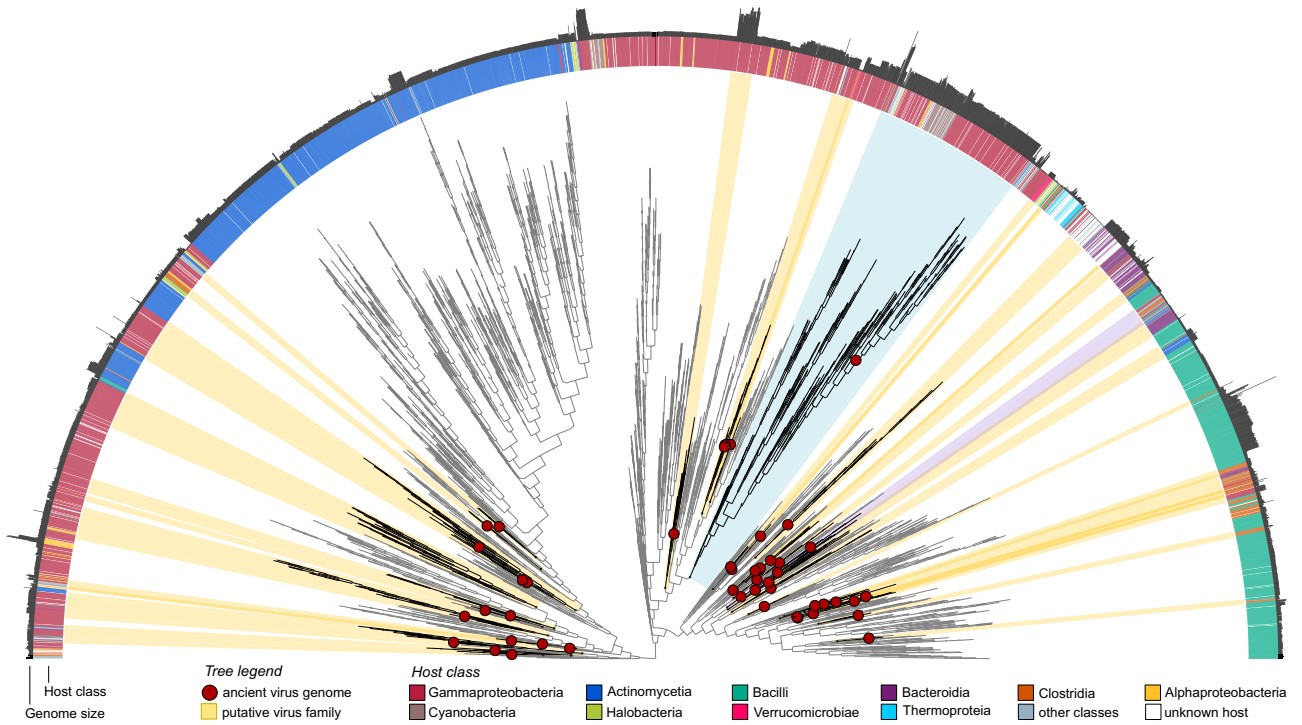

**Fig. 3 | Proteomic tree of ancient and contemporary phage genomes.** A tree generated in ViPTreeGen[70] encompassing 49 high-quality and complete ancient phages, their 265 closest relatives from IMG/VR[29], and all 4703 prokaryote-infecting viruses from RefSeq[76]. The ancient phages are identified with red circles. Putative families were defined using a percentage of shared genes and amino acid identity between each pair of viruses. Clusters containing at least one ancient phage are highlighted in yellow and listed in Supplementary Data 11. The purple and blue highlighted clades represent putative families of *Salmonella*-infecting phages and T4 bacteriophage-related genomes, respectively. Outer rings provide additional information for each phage, such as its host assignment and genome size. Source data are provided as a Source Data file.

We also searched for ancient relatives of modern Crassvirales that are widespread in mammalian intestinal viromes. We found four such sequences. The two longest contigs are probably fragments of the same genome, similar to *Bacteroides* phage PhiCrAssBcn21 (Supplementary Fig. 4a). The remaining sequences are shorter (21 kb and 42 kb) and bear little resemblance to modern phages (less than 15% ANI; Supplementary Data S6 and Supplementary Fig. 4b).

Notably, our taxonomic analysis of high-quality ancient genomes was limited by the database-dependent tool CheckV, which has a bias for phages similar to known reference genomes. Consequently, the other 249 aMGVs could still include viral genomes unlike any sequence characterised to date. Despite this limitation, we have shown that our approach is promising for discovering previously unknown viruses and shedding light on dark matter lingering in ancient metagenomes.

### Persistence of *Mushuvirus mushu*

Among the collection of ancient phages, we found the genome sequence of a bacteriophage from the *Mushuvirus mushu* species recently discovered in a prophage analysis of its host, *Faecalibacterium prausnitzii*[33]. In that study, the prophage could not be induced after DNA damage, but virions were observed in the gastrointestinal tract in a mouse model[33]. We believe that our aMGV represents an ancient sequence because the frequency of C → T substitutions at the first position of *Mushuvirus mushu* reads was high (0.042) compared to other sequences in our collection of ancient viruses (mean: 0.025; see Supplementary Fig. 5 and Supplementary Data 3). Moreover, the *F. prausnitzii* host of *Mushuvirus mushu* is a strict anaerobe that typically inhabits the mucosal surface in the gastrointestinal tract[34], so it is unlikely it could have contaminated the original palaeofaeces sample[21].

The ancient *Mushuvirus mushu* genome that was recovered from 1300-year-old faecal material from the Zape cave in Mexico was 97.7% identical to the modern reference genome FP_Mushu (RefSeq accession: NC_047913.1) that was extracted from wastewater in France[33] (Fig. 4a, b). The mean sequence identity to ten IMG/VR genomes from the same species that were derived from the large intestine was 97.1% ± 0.6% (Fig. 4c and Supplementary Data 12). The ancient and modern genome sequences had similar lengths (36,623 bp and 36,636 bp, respectively) and perfect colinearity between 52 protein-coding genes (Fig. 4b). Phage mutation rates that are reported in the literature range from $1.976 \times 10^{-4}$ to $4.690 \times 10^{-3}$ nucleotide alterations per site per year due to substitutions and recombinations for virulent phages[7,35], and $1.154 \times 10^{-4}$ substitutions per site per year for temperate phages[6]. As shown in Fig. 4d, the probability of finding a 36 kbp long genome sequence with at least 97.7% identity approaches zero after a little over 200 years, even at the lowest mutation rates reported in the literature[6,7,35]. Notably, this calculation assumes direct ancestral relationship between both viruses, as encountering such similar genomes in two sister clades that accumulate mutations independently is even less likely. Surprised by such a low mutation rate, we estimated intra-population genetic diversity (microdiversity)[36] of *Mushuvirus mushu* in modern Hadza hunter-gather's gut[37]. However, we found that both nucleotide diversity and the number of divergent sites of *Mushuvirus* were higher than the median for other phage sequences (Fig. 4e and Supplementary Data 13). Thus, low mutation rates are unlikely to explain the observation of two nearly identical *Mushuvirus mushu* genomes in the span of 1300 years.

A total of 22,059 sequencing reads were mapped to the ancient *Mushuvirus mushu* genome, resulting in 28x genome coverage. Our comparison of the ancient and modern genomes revealed 869 single nucleotide variants (SNVs). The distribution of these SNVs was concentrated in the specific section of the gene encoding the Hoc-like capsid decoration protein (Fig. 4b). This fragment is the target of a diversity-generating retroelement (DGR) that produces a large number of localised mutations through error-prone reverse transcription and

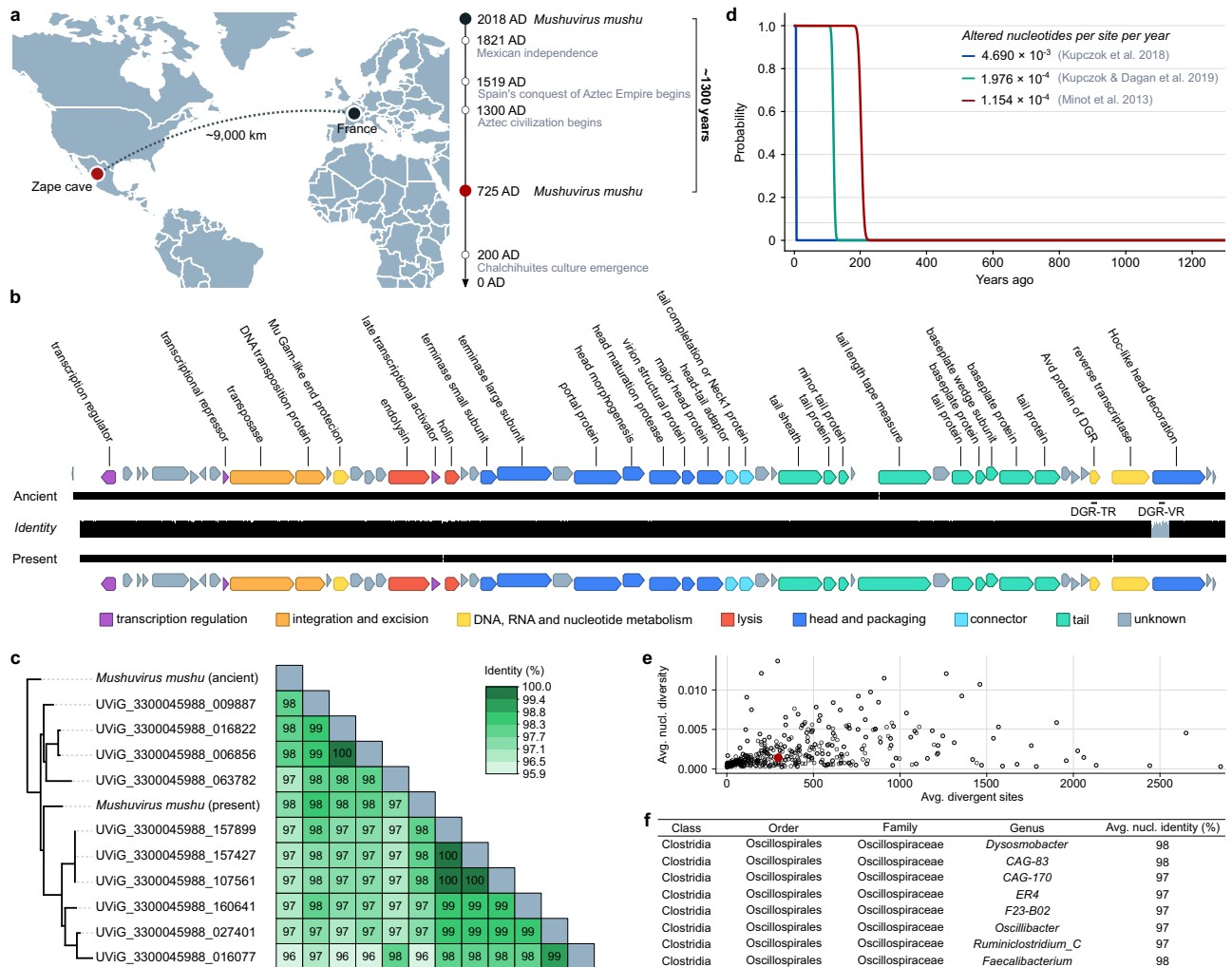

**Fig. 4 | Stability of the *Mushuvirus mushu* prophage genome. a** Geographic and historical distances between the modern reference (France) and the ancient *Mushuvirus mushu* (from the Zape cave in Mexico) genomes. **b** Comparison of sequence conservation and gene organisation between the ancient and modern reference (NC_047913.1) genomes of *Mushuvirus mushu*. Black lines indicate alignments with small gaps. The height of the middle plot depends on the identity between genomes. The lowest similarities (grey colour) were concentrated in a ∼ 600-nucleotide variability region of the Hoc-like gene affected by the diversity-generating retroelement (DGR) mechanism. Two major structural differences in the ancient and modern genomes constituted an additional codon stop in a gene of unknown function (genome position: 9518) and 17 nucleotide insertion/deletions, which changed the open reading frame in the tail length tape measure protein. **c** Intergenomic similarity matrix calculated by VIRIDIC[63] revealed high identity between ancient and modern representatives of the species *Mushuvirus mushu*

deposited in RefSeq and IMG/VR v.4. **d** Probability of a 36,630-nucleotide phage genome having at least 97.7% unaltered nucleotides over a period of 1300 years. The probability was calculated using a binomial cumulative distribution function assuming different mutation rates reported in the literature[6,7,35]. The blue and green lines represent the mutation rates due to substitutions and recombinations in lytic phages, with rates of $4.690 \times 10^{-3}$ and $1.976 \times 10^{-4}$ altered nucleotides per site per year, respectively. The green line includes a rate of $1.154 \times 10^{-4}$ altered nucleotides per site per year due to substitutions in temperate phage genomes. **e** Intra-population genetic diversity (microdiversity) of *Mushuvirus mushu* (red point) and other bacteriophages in modern Hadza hunter-gather's gut[37] based on calculation from InStrain[36]. **f** Table with *Mushuvirus mushu* hosts found using the BLASTn search against the Unified Human Gastrointestinal Genome collection[45] and the Genome Taxonomy Database[44]. Source data are provided as a Source Data file.

is involved in host switching in some phages[38–40]. Indeed, the Hoc-like gene had a lower level of AAI to the reference FP_Mushu (91.8%) than other protein-coding genes (mean identity: 98.22%) and accounted for 16% of all SNVs in the genome. Notably, according to recently published RNA-seq data from cultured strains of *F. prausnitzii*, the Hoc-like gene is expressed at levels similar to a typical bacterial gene under most studied conditions[41] (Supplementary Fig. 6). This finding suggests that the Hoc-like protein might play a role in lysogenized bacterial cells. Despite the potential significance of the Hoc-like gene in the evolution of *Mushuvirus mushu*, its function remains poorly understood with only homologues of the gene being characterised as enhancing binding to mucus on the mucosal surface of the intestine and potentially increasing encounters with mucosa-dwelling hosts[42,43].

To analyse whether the ancient *Mushuvirus mushu* sequence represented an integrated prophage, we investigated the flanking regions of the contig (see "Methods" section). These regions differed between the ancient and modern viruses indicating a potential bacterial origin. The sequence similarity of the flanking regions of ancient *Mushuvirus mushu* points to *Oscillospiraceae* genus ER4, suggesting this may be a host for the ancient *Mushuvirus mushu* (Supplementary Data 13). *Mushuvirus mushu* appears to have a broad host range[33]; when comparing the ancient genome to bacterial sequences from the Genome Taxonomy Database (GTDB)[44] and the Unified Human Gastrointestinal Genome (UHGG) collection[45], we found prophage genomes with 97% sequence identity in seven different *Oscillospiraceae* genera, including *Faecalibacterium* (see Fig. 4f and Supplementary Data 15).

## Discussion

In this study, we demonstrated the large-scale reconstruction of high-quality ancient phage genomes using state-of-the-art bioinformatic methods. To validate the authenticity of the reconstructed ancient genomes, we analysed their DNA damage patterns, relationships with contemporary viruses, and their host associations. There were no significant similarities to known viruses for approximately half of the reconstructed genomes. This indicates that our understanding of human gut virome history is limited. Nevertheless, advancements in viral metagenomics and access to well-preserved aDNA samples hold promise for expanding our understanding of virosphere evolution.

Bacteriophage genomes are often variable − shaped by rampant mutations, recombinations, and horizontal gene transfers[1,7]. Therefore, recent reports of near-identical phage genomes found over vast geographical distances[46] and spanning several years in the same location[47] were striking. Here, we reconstructed a high-quality ancient genome sequence of *Mushuvirus mushu* that was highly conserved despite at least 1300 years of evolution and a presence on different continents.

We propose three hypotheses pertaining to the unique preservation of the *Mushuvirus mushu* genome. First, the high conservation may arise from its replication strategy. The phage displays genomic characteristics similar to transposable "Mu-like" phages. Typically, these phages are associated with extensive rearrangement of the host's genetic material and highly variable genome termini that result from prophage excision (the Mu abbreviation refers to mutator). However, transposable bacteriophages like *Mushuvirus mushu* lack their own replicase enzyme, relying instead on the host's polymerase III for DNA replication[48]. This enzyme tends to have error rates orders of magnitude lower than these typical phage polymerases. Notably, this hypothesis seems to be at odds with our estimations of intra-population genetic diversity of phages from the gut of modern hunter-gatherers that indicated a relatively high microdiversity of *Mushuvirus mushu*.

The second hypothesis relates to the phage's extensive host range, encompassing multiple *Oscillospiraceae* species. These species are prevalent in human populations ranging from hunter-gatherers to industrialised societies[37]. Thus, unrestricted genetic drift has the potential to disrupt specific adaptations for individual hosts, but this risk is counteracted by purifying selection. This mechanism may help to create the evolutionary stasis of viruses in long-term host relationships[49] and results in obscuring molecular dating of viral lineages based solely on modern genomes[50].

The third, perhaps most likely explanation for the conservation of *Mushuvirus mushu* is that it has existed mostly as an integrated prophage, and hence part of its evolution was similar to that of the host genome. Indeed, the identity between modern genomes of *Faecalibacteria* and their ancient counterparts found in human palaeofaeces (95%-97% ANI retained after 1000–2000 years)[17] is comparable to this observed of modern and ancient strains of *Mushuvirus*. Some prophages of sporulating bacteria represent relics of a bygone era, that survived and spread by persisting within endospores and re-emerging in a relatively unchanged form[51]. However, none of the predicted hosts of *Mushuvirus mushu* have been reported to form endospores. As these hypotheses are not mutually exclusive, the observed conservation of the *Mushuvirus mushu* genome might result from different combinations of factors like replication strategy, genetic constraints imposed by the broad host range, and/or dormancy of the virus. Finally, there might be other reasons that we have not yet considered.

Regardless of the specific mechanism underlying the high conservation of *Mushuvirus mushu* genome, recent studies are increasingly reporting ancient phages that bear a remarkable resemblance to their modern counterparts. For instance, a recent preprint study on 2-million-year-old microbial and viral communities in North Greenland discovered three phage genomes that had high damage patterns and showed average nucleotide identity exceeding 96% when compared to contemporary phages[52].

To sum up, our study highlights the utility of publicly accessible ancient metagenomes in investigating viruses associated with microorganisms. Our focus was on sequences from well-preserved gut and palaeofaeces samples. However, much of the currently available data comes from ancient teeth or dental plaque, and these samples await further exploration. In the future, we anticipate that similar studies of virome in ancient metagenomes will contribute to elucidating the complex history of viruses and their role in co-evolving with bacterial, animal and plant hosts.

## Methods

### Sample selection

Based on community-curated metadata from the AncientMetagenomeDir[53], we selected 72 aDNA metagenomic libraries from 30 well-preserved human faeces or digestive gut contents. Data from palaeofaeces primarily originated from archaeological excavations in caves located in Boomerang Shelter, Arid West Cave or Zape in the USA and Mexico ($n = 38$) as well as the underground salt mines of the Hallstatt in Austria ($n = 19$). Moreover, we used the digestive contents ($n = 15$) from a biopsy of the Tyrolean Iceman mummy discovered more than 30 years ago in a melting glacier[19]. The samples represented material from 150 to 5300 years ago. Details about original publications, localisations (coordinates), and samples (sequencing depth and instruments) are described in the supplementary materials (Supplementary Data 1).

### Sequence data preprocessing

Pair-ended Illumina reads were trimmed using Cutadapt v.4.1[54] with a quality cutoff = 25, minimum read-length = 30, and a minimum overlap with adapter sequence to reads = 1. A total of 2,628,045,312 clean reads from all 72 libraries were mapped to the *Homo sapiens* reference genome (hg37) using KneadData v.0.12.0 (with --bypass-trim option) to filter out human DNA (https://github.com/biobakery/kneaddata). The quality of the 2,352,455,887 remaining reads after preprocessing was controlled in Fastqc v.0.12.0[55].

### De novo contig assembly

Filtered reads from each library were assembled into contigs using Metaspades v.3.15.5[56] with default settings. Only contigs longer than 4000 nucleotides with a minimum coverage of 20 were considered in the downstream analyses (Fig. 1).

### aDNA authentication

After preprocessing, clean reads were mapped to the assembled contigs using Bowtie2 v.2.4.4[57] with default settings. It was observed that these default settings did not significantly affect the alignment compared to the *--sensitive* settings, likely due to the low aDNA damage (Supplementary Fig. 7). The resulting alignment was sorted and indexed with SAMtools v.1.14[58]. DNA damage patterns observed in reads were used to label corresponding contigs as ancient or modern. The analysis was run using PyDamage v.0.70[15] - a programme that calculates the frequency of C to T transitions at the first 20 positions of mapped reads compared to a reference sequence. The filtering threshold for predicted accuracy was determined by the Kneedle method[59] and we imposed an additional cut-off of transition frequency (minimum 0.01 at the first position) to filter out contigs with weakly damaged reads that could introduce a random noise generated by the inherent error of the Illumina method. Additionally, the same analytical process of authentication was performed for 943 randomly selected contigs from the Metagenomic Gut Viruses (MGV) database[31]

to compare deamination patterns between modern and ancient viral genomes (Fig. 1c, d).

## Viral contigs identification

Three machine learning tools were used to identify viral contigs. The first was Jaeger v.1.1.0, a deep-learning model that identifies phage genome sequences in metagenomes (https://github.com/Yasas1994/Jaeger) based on automatic compositional feature extraction. The second and third were VIBRANT v.1.2.1[24] and VirSorter2 v.2.2.3[25], which rely on analysing HMM profiles representing conserved families and/or domains similar to predicted proteins but applying different classifiers and reference databases. Jaeger and VIBRANT were run with default settings. For VirSorter2, we used the positional arguments '--include-groups dsDNAphage,NCLDV,ssDNA,lavidaviridae all'. Contigs classified as viral by at least two tools were further analysed.

## Bacteriophage lifestyle prediction

Bacteriophage lifestyles were predicted based on the presence of similar prophages in bacterial genomes and lysogeny-associated protein domains. Prophages were defined as BLASTn v.2.13.0+ hits against UHGG collection[45] and GTDB[44] with minimum 50% coverage of the aMGV (only genomes longer than 5000 nucleotides were used as a query). To predict lifestyle based on domain content we used BACPHLIP v.0.9.6[60]. We classified bacteriophages as temperate, if at least one method indicated this lifestyle, other genomes were considered as virulent.

## Genomes quality assessment

CheckV v.1.0.1[61] was applied to assess the genome quality of ancient viral contigs using the 'end_to_end' command. Ancient viral genomes classified as complete, high-quality, medium-quality, or fragments longer than 20 kb and with at least one viral gene were considered in the next steps.

## Viral contigs clustering

The ancient MGVs (n = 383) were clustered into 298 species-level viral operational taxonomic units (vOTUs) using scripts, published in the CheckV repository (https://bitbucket.org/berkeleylab/checkv/src/master/). Accordingly, sequences were grouped based on 95% ANI and 85% alignment fraction of the shorter sequence, as recommended in MIUViG (Minimum information about an uncultivated virus genome)[26].

## Gene-sharing network

We selected 10 modern phage genomes for each aMGV from the IMG/VR (v.4 - high-confidence genomes only) to compare ancient phages with their modern counterparts from different environments. Specifically, we selected genomes with the highest number of shared proteins determined by a DIAMOND v.2.0.15[62] search in the 'blastp' mode (--very-sensitive) with a minimum of 50% query coverage and 50% sequence identity. We then visualised this collection of aMGVs, selected modern viral genomes, and all prokaryotic DNA viruses with assigned ICTV taxonomy (VMR_20-190822_MSL37.2, created 08/31/2022) using vConTACT2 v.0.11.3[28]. The network was displayed in Cytoscape v.3.9.0 and refined in Inkscape v.1.2.2.

## Comparison of aMGVs to contemporary bacteriophage genomes

The genomic sequences of 298 aMGVs were queried in the BLASTn searches against genomes of contemporary viruses from IMG/VR v.4, GenBank, and RefSeq. For each aMGV, we selected the top 30 contemporary virus genomes with the highest BLASTn alignment score and calculated ANI and AAI between the query aMGV and the selected genomes using VIRIDIC v.1.1[63] and EzAAI v.1.2.2[64], respectively. The contemporary phage with the highest ANI was identified as the closest known modern relative to each aMGV.

## Host prediction

Four computational tools (BLASTn[65], PHIST[66], VirHostMatcher-Net[67], and RaFAH[68]) were used to assign hosts to ancient phages and sequences from IMG/VR v.4 (only PHIST). PHIST v.1.1.0 and BLASTn v.2.13.0+ predictions were run against representative genomes of GTDB[44] database v.07-RS207 (62,291 bacterial species + 3,412 archaeal species). Prokaryotic species whose genomes obtained the highest similarity score to the virus genome and had an e-value < 10^-5 (BLASTn) or p-value < 10^-5 (PHIST) were assigned as a putative host. For methods integrating machine learning approaches such as VirHostMatcher-Net v.1.0 or RaFAH v.0.1, we selected host predictions with, at minimum, 0.5 and 0.14 scores, respectively.

## Taxonomy assignment, clustering, and phylogenetic analysis

Taxonomic assignment of viral genomes was performed in the geNomad v.1.3.3 tool[32] using the 'annotate' function. To classify aMGVs at the genus and family level, viral genomes were clustered using a combination of gene sharing and AAI. Initially, we selected aMGVs assessed as high-quality or complete by CheckV and then added prokaryotic viruses from RefSeq (n = 4703; access: 30.01.2023) and IMG/VR sequences (n = 265) to form clusters (VC) with aMGVs (see: Methods, Gene-sharing-network). In this collection, pairwise protein sequence alignments were performed using DIAMOND[62] with the options '-e-value 1 × 10 − 5−max-target-seqs 10000'. Next, we calculated the percentage of shared genes and AAI between each pair of viruses. Following the criteria from previous studies[31,69], we kept connections between viruses with >20% AAI and >10% genes shared for the family level and >50% AAI and >20% genes shared for the genus level. Finally, clustering was performed based on the connections between viral genomes using MCL with the option '-I 1.2' for the family level or '-I 2' for the genus level. All scripts used to perform analyses at this step are available at (https://github.com/snayfach/MGV/blob/master/aai_cluster/README.md). To visualise the phylogenetic relationships of genus- and family-level groups, we generated a proteomic tree of 5017 viral sequences using ViPTreeGen v.1.1.3[70] and GraPhlAn v.1.1.3[71].

## Analyses of the *Mushuvirus mushu* genome

We performed a BLASTn search of 298 ancient metagenomic viral sequences against the nr/nt NCBI database. This search revealed that one identified aMGV (NODE_310_length_36983_cov_28.516681) was remarkably similar to present-day *Mushuvirus mushu* (NC_047913.1). This genome was present in a vContact2 cluster, along with one reference from NCBI (contemporary genome of *Mushuvirus mushu*) and 10 vOTUs from IMG/VR v.4 (high-confidence genomes only). All 12 sequences from the vContact2 cluster were aligned using MAFFT v.7.308 to differentiate the core of the phage genome (~36,623 bp) from the flanking regions coming from host integration sites (~241 bp and ~119 bp in the ancient contig). Only those core sequences were used in further analyses. To assess the nucleotide variability of modern and ancient *Mushuvirus mushu* sequences, we calculated the intergenomic similarity within the cluster using the VIRIDIC[63] with the following parameters: '-word_size 7 -reward 2 -penalty −3 -gapopen 5 -gapextend 2'. A phylogenetic tree (Fig. 4d) was constructed with the obtained similarity matrix using the bioNJ algorithm with default parameters[72]. To annotate protein-coding genes in the analysed genomes, we used an end-to-end script (https://github.com/Yasas1994/phage_contig_annotator) that annotates phage genes based on the HMMER v.3.3.2 search against Prokaryotic Virus Remote Homologous Groups[73]. Visualisation and manual curation of the genome were conducted in Geneious Prime v.2023.04 (Fig. 4c). The MAFFT plugin for the same tool was used to generate multiple sequence alignments of entire viral genomes. The identity of protein-coding genes at the amino acid level was calculated based on local alignment performed using EMBOSS v.6.6.0.0[74]. To calculate SNVs in the ancient genome, a Python script was used (https://github.com/pinbo/msa2snp). We

detected template and variable regions of DGRs using myDGR[38]. Finally, we assigned potential hosts by searching for sequences similar to the prophage core and flanking regions (blastn -task megablast) in the UHGG collection[45] and GTDB[44]. We measured the population microdiversity of *Mushuvirus mushu* in modern hunter gathers' gut (Fig. 4e) by mapping raw sequences to the complete bacteriophage genomes from samples where *Mushuvirus mushu* was previously detected[37] using Bowtie2 with default settings. Next, we calculated average nucleotide diversity and average divergent sites using InStrain v.1.8.0[36].

## Reporting summary

Further information on research design is available in the Nature Portfolio Reporting Summary linked to this article.

## Data availability

Ancient phage genome sequences and their gene annotations generated in this study have been deposited in the Zenodo database [https://doi.org/10.5281/zenodo.7919433]. The reconstructed ancient genome sequence of *Mushuvirus mushu* is available from NCBI GenBank under accession BK063464. Supporting data generated in this study are provided in the Supplementary Information, Source Data and Supplementary Data files. Accession numbers of ancient metagenomic samples used in this study are available in the AncientMetagenomeDir [https://github.com/SPAAM-community/AncientMetagenomeDir]. Other databases used in the study include: IMG/VR v.4 [https://genome.jgi.doe.gov/portal/IMG_VR/], GTDB v.07-RS207 [https://data.ace.uq.edu.au/public/gtdb/data/releases/release207/], UHGG v.2.0 [http://ftp.ebi.ac.uk/pub/databases/metagenomics/mgnify_genomes/human-gut/v2.0/], NCBI GenBank release 251 [https://www.ncbi.nlm.nih.gov/genbank/] and NCBI RefSeq release 215 [https://www.ncbi.nlm.nih.gov/refseq/], PHROG v.4 [https://phrogs.lmge.uca.fr], and Virus Metadata Resource release 12/02/2022 from ICTV [https://ictv.global/vmr]. Source data are provided with this paper.

## Code availability

Bioinformatic scripts and a guide to the data analysis performed in this study are provided in the GitHub repository (https://github.com/rozwalak/Ancient_gut_phages)[75].

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

## Acknowledgements

This work was funded by the Polish Ministry of Science and Higher Education under the programme "Perły Nauki", project number PN/01/0063/2022. The total value of the project 228,448PLN was awarded to P.R.; A.Z is supported by Polish National Science Centre [2018/31/D/NZ2/00108], and B.E.D. is supported by the European Research Council (ERC) Consolidator grant 865694: DiversiPHI, the Deutsche Forschungsgemeinschaft (DFG, German Research Foundation) under

Germany´s Excellence Strategy – EXC 2051 – Project-ID 390713860, the Alexander von Humboldt Foundation in the context of an Alexander von Humboldt-Professorship founded by the German Federal Ministry of Education and Research. B.E.D and Y.W. are supported by the European Union's Horizon 2020 research and innovation program, under the Marie Skłodowska-Curie Actions Innovative Training Networks grant agreement no. 955974 (VIROINF). The computations were performed at the PLGrid Infrastructure and the Poznan Supercomputing and Networking Center (grant pl0074-02 and grant pl0243-01).

## Author contributions

P.R. conceived the idea of studying ancient phages; P.R., A.Z., J.B. and B.E.D. designed the research; P.R. and A.Z. selected samples and performed preprocessing, assembly and aDNA authentication; J.B. and Y.W. performed identification of viral contigs; P.R. and A.Z. created a gene-sharing network and performed host prediction; P.R., A.Z. and J.B. performed taxonomy assignment, clustering, phylogenetic studies, and the analyses of the Mushuvirus mushu genome; Y.W. contributed to phage genome annotation; P.R., A.Z., J.B., and B.E.D. analysed the results. P.R. wrote the manuscript with substantial contributions from A.Z., J.B. and B.E.D. All authors reviewed and approved the manuscript.

## Competing interests

The authors declare no competing interests.
