## [Peer Review File · Nature Communications]

Ultraconserved bacteriophage genome sequence identified in 1300-year-old human palaeofaecesReviewer #1 (Remarks to the Author):

This is a novel study demonstrating the feasibility of analysing ancient phage genomes from palaeofaeces. The methodology appears sound and convincing arguments are given that these are ancient phage sequences which are up to 5000 years old, potentially giving unique insights into ancient human gut phages. The manuscript focuses in on one particular phage, Mushuvirus mushu, with the main finding being that this ancient phage (1200 years) has a genome 97% identical to the modern variant. This is a very interesting finding given the known rapid evolution of phages.

My main feedback on the manuscript is that it seems incomplete, as the reader is left wondering about the other (~300) phages. The authors have put a lot of effort into carefully identifying, grouping and predicting hosts for this valuable dataset of ancient gut phages, analysed one phage in detail and not analysed the others in any further way. It is mentioned that approximately half were not related to modern variants, so half were (as shown in Fig 2). It's therefore difficult to know if the take home message is 'Ultraconserved bacteriophages...' as in the title, or 'Ancient phages are all different to modern variants (except one)', or something else.

To improve the manuscript, I recommend the authors at least describe and tabulate the other phages in a similar way. E.g. how similar were the contigs found to modern variants in GenBank or IMG/VR (I couldn't get this information from Fig 2). Apologies if this is already given, but I couldn't find it in the supplements. It would also be good to see a couple more genome comparisons for different phages. In particular, I'm thinking of CrAssphages, which are near universal in human guts, often making up the majority of viral reads. For example, I picked a random CrAssphage contig from the dataset 'NODE_28_length_59943_cov_42.907260', (1200 years old) and blasted 4 genes from this, I get 97% 98.5%, 93% ,95% identity to CrAssphages in GenBank (blastp), and better hits from IMGVR. If I align a few of the matching IMGVR genomes with this ancient variant, it shows they are about 80% ANI (core regions are 90% identical). A genome comparison could be made here also. Such additional information for some other phages would help complete the story.

Regarding the Mushu phage, a little more discussion here would also help the reader. Is there something about the lifestyle of this one phage that makes it particularly stable? All the modern variants in IMG/VR also seem to be at a similar ANI (Figure 4c). Could the authors discuss it a little further. This might be a particularity unusual phage given it was predicted to replicate by transposition in ref 33.

Reviewer #2 (Remarks to the Author):

What are the noteworthy results?

The discovery of an ancient genome nearly identical to a present-day bacteriophage known as Mushuvirus mushu is a significant finding. The ancient genome shares 97.7% nucleotide identity with its modern counterpart, despite 1300 years of evolution, indicating a long-term relationship between the prophage and its host. This discovery provides a unique perspective on the stability of some bacteriophage genomes over extensive periods. The successful de novo reconstruction and authentication of 297 other phage genomes from the last 5300 years, including those from unknown families, is also a significant achievement.

Will the work be of significance to the field and related fields? How does it compare to the established literature? If the work is not original, please provide relevant references.

This work presents, to my knowledge, of of the first (if not the first) application of reference free metagenomics (*using de novo assembly*) to ancient DNA bacteriophages.

The authors use recently released tools to re-analyze published data and search for DNA bacteriophages, which is in itself a novelty in the field of ancient DNA.

Does the work support the conclusions and claims, or is additional evidence needed?

Regarding the "catalogue of ancient phages", the authors have done a good job of applying an ensemble approach, by using different viral sequence identification tools.

The gene network analysis presented here is an interesting and relatively convincing way of tackling the mammalian host/ecosystem identification issue.

The authors could however go a bit further in the catalogue, by investigating whether these phages are all integrated in their bacterial hosts as prophages, or were free living as lytic phages in the gut microbiomes (a simplified approach of what they've done for the later section)

Regarding the reconstruction of the *Mushuvirus mushu* phage genome, the figure 4e seems out of place, and as it comes from unrelated work, and the mention of the findings and a citation to the original work should suffice in the main text.

The methodology is nevertheless sound.

Are there any flaws in the data analysis, interpretation and conclusions? Do these prohibit publication or require revision?

The very minimal DNA damage encountered by the authors is most likely unfortunately explained by a small yet important methodological flaw. The authors have used bowtie2 in default mode, however, for ancient DNA, it is highly recommended to use its very sensitive settings, and even more importantly, allow for mismatches at the seeding step (setting -N 1). Using the default setting can unfortunately discard reads carrying ancient DNA damage at the seeding step, ultimately lowering the aDNA damage.

Is the methodology sound? Does the work meet the expected standards in your field?

This research is pioneering a new thematic of the field, so the expected standards are up to discussion. However, comparing to ancient microbial reconstruction standards, this work would definitely the standards of the field (except for the above mentioned methodological issue)

Is there enough detail provided in the methods for the work to be reproduced?

Almost ! The methods are described in the main text, and the authors have made their code their analysis script available online. However, they should pin the versions of the different softwares in the conda environments of their snakemake pipeline to ensure the reproducibility.

Reviewer #3 (Remarks to the Author):

The manuscript by Rozwalak et al presents the results of one of the first metagenomic studies of the ancient human gut virome. The authors used state of the art methods in order to extract, taxonomically classify and annotate viral contigs from the total community metagenome. The use of deamination data and the enrichment of sample for phages related to the mammalian (mostly human) gut phages, convincingly shows that the recovered genomes indeed belong to the ancient gut phages, and not to environmental phages contaminating the sample. High conservation level of Mushuvirus mushu infecting *F. prausnitzii* (one of the main species of the human gut commensals), is interesting and noteworthy.

I believe this manuscript is of interest to the broad audience and is publishable in its present form.

I have only a couple of small comments:

Lines 207-208. Do authors assume here that the ancient *Mushuvirus mushu* is a direct predecessor of the modern virus? Given the diversity of gut phages, it is equally possible, if not more likely, that it is in fact a sister clade (different subspecies).

Lines 235-236: Family *Rumicococcaceae* is an older, deprecated name of the family *Oscillospiraceae* (<https://lpsn.dsmz.de/family/oscillospiraceae>)

Response to Reviewer 1

- C:** This is a novel study demonstrating the feasibility of analysing ancient phage genomes from palaeofaeces. The methodology appears sound and convincing arguments are given that these are ancient phage sequences which are up to 5000 years old, potentially giving unique insights into ancient human gut phages. The manuscript focuses in on one particular phage, Mushuvirus mushu, with the main finding being that this ancient phage (1200 years) has a genome 97% identical to the modern variant. This is a very interesting finding given the known rapid evolution of phages.
- R:** We are grateful for the Reviewer's kind words and thoughtful evaluation of our manuscript.
- C:** My main feedback on the manuscript is that it seems incomplete, as the reader is left wondering about the other (~300) phages. The authors have put a lot of effort into carefully identifying, grouping and predicting hosts for this valuable dataset of ancient gut phages, analysed one phage in detail and not analysed the others in any further way. It is mentioned that approximately half were not related to modern variants, so half were (as shown in Fig 2). It's therefore difficult to know if the take home message is 'Ultraconserved bacteriophages...' as in the title, or 'Ancient phages are all different to modern variants (except one)', or something else. To improve the manuscript, I recommend the authors at least describe and tabulate the other phages in a similar way. E.g. how similar were the contigs found to modern variants in GenBank or IMG/VR (I couldn't get this information from Fig 2). Apologies if this is already given, but I couldn't find it in the supplements.
- R:** Thank you for pointing out this issue - we agree that the description of the remaining phages has been somewhat cursory. In the revised manuscript, we have provided additional information for all 298 phages, including: (i) their ANI and AAI similarities to modern genomes from GenBank, RefSeq, and IMG/VR (refer to lines: 126-128, Supplementary Tables 6 and 7, Supplementary Fig. 1), (ii) prediction of their lifestyle (lines: 104-105, Supplementary Table 4), and (iii) gene annotation presented in separate GFF files (accessible via Zenodo).
- C:** It would also be good to see a couple more genome comparisons for different phages. In particular, I'm thinking of CrAssphages, which are near universal in human guts, often making up the majority of viral reads. For example, I picked a random CrAssphage contig from the dataset 'NODE_28_length_59943_cov_42.907260', (1200 years old) and blasted 4 genes from this, I get 97% 98.5%, 93% ,95% identity to CrAssphages in GenBank (blastp), and better hits from IMGVR. If I align a few of the matching IMGVR genomes with this ancient variant, it shows they are about 80% ANI (core regions are 90% identical). A genome comparison could be made here also. Such additional information for some other phages would help complete the story.
- R:** We agree that additional genome comparisons of modern and ancient genomes may be

interesting, and we have added relevant supplementary figures for the ancient T4-like and four CrAss-like phages. In the process, we discovered that the two longest contigs from the *Crassvirales* cluster are probably fragments of the same virus and, taken together, reconstitute a nearly complete genome. New genome organization maps and comparisons with closest modern relatives are now presented in Supplementary Figures S3 and S4. Detailed information about ANI, AAI and coverage of associated alignments are provided in Supplementary Tables 6 and 7. All these analyses are described in lines 168-170 and 183-188.

- C:** Regarding the Mushu phage, a little more discussion here would also help the reader. Is there something about the lifestyle of this one phage that makes it particularly stable? All the modern variants in IMG/VR also seem to be at a similar ANI (Figure 4c). Could the authors discuss it a little further. This might be a particularity unusual phage given it was predicted to replicate by transposition in ref 33.
- R:** Thank you very much for this suggestion. We admit that our initial discussion regarding the stability of the *Mushuvirus mushu* genome was rather rudimentary. In the revised manuscript, we have added a new paragraph in Discussion (lines: 277-315), outlining three different hypotheses that might explain the high conservation of the phage genome.

Response to Reviewer 2

- C:** The discovery of an ancient genome nearly identical to a present-day bacteriophage known as *Mushuvirus mushu* is a significant finding. The ancient genome shares 97.7% nucleotide identity with its modern counterpart, despite 1300 years of evolution, indicating a long-term relationship between the prophage and its host. This discovery provides a unique perspective on the stability of some bacteriophage genomes over extensive periods. The successful de novo reconstruction and authentication of 297 other phage genomes from the last 5300 years, including those from unknown families, is also a significant achievement. This work presents, to my knowledge, of the first (if not the first) application of reference free metagenomics (*using de novo assembly*) to ancient DNA bacteriophages. The authors use recently released tools to re-analyze published data and search for DNA bacteriophages, which is in itself a novelty in the field of ancient DNA.
- R:** We would like to thank the Reviewer for the positive assessment of our work and thorough review.
- C:** Regarding the "catalogue of ancient phages", the authors have done a good job of applying an ensemble approach, by using different viral sequence identification tools. The gene network analysis presented here is an interesting and relatively convincing way of tackling the mammalian host/ecosystem identification issue. The authors could however go a bit further in the catalogue, by investigating whether these phages are all integrated in their bacterial hosts as prophages, or were free living as lytic phages in the gut microbiomes (a simplified approach of what they've done for the later section).
- R:** Thank you for your suggestion. In the revised manuscript, we have included details regarding the predicted lifestyle of ancient phages (lines 104-105, 383-391, and Supplementary Table 4). To summarize, among the 298 ancient phages, 122 (41%) were predicted to be temperate. This prediction was made by identifying genes associated with the lysogenic lifestyle or by detecting similar phage genomes within the genomes of human gut bacteria (with at least 50% of the phage genome length aligning with the host genome). Conversely, the remaining 176 ancient phages (59%) did not satisfy either of these criteria and, consequently, were presumed as lytic.
- C:** Regarding the reconstruction of the *Mushuvirus mushu* phage genome, the figure 4e seems out of place, and as it comes from unrelated work, and the mention of the findings and a citation to the original work should suffice in the main text.
- R:** We agree - we have relocated Figure 4e to Supplementary Figure 6.
- C:** The very minimal DNA damage encountered by the authors is most likely unfortunately explained by a small yet important methodological flaw. The authors have used bowtie2 in default mode, however, for ancient DNA, it is highly recommended to use its very sensitive

settings, and even more importantly, allow for mismatches at the seeding step (setting -N 1). Using the default setting can unfortunately discard reads carrying ancient DNA damage at the seeding step, ultimately lowering the aDNA damage.

R: Following the Reviewer's suggestion, we compared DNA damage of ancient phage genomes between two sets of Bowtie2 results: one using default parameters and the other employing the sensitive parameters recommended by the reviewer. We did not find significant differences in C-to-T transition frequencies between the two Bowtie2 modes. The results of this comparison are now described in the Methods section (lines 357-359) and presented in Supplementary Figure 6.

C: This research is pioneering a new thematic of the field, so the expected standards are up to discussion. However, comparing to ancient microbial reconstruction standards, this work would definitely the standards of the field (except for the above mentioned methodological issue).

R: We are honored to receive such a favorable opinion. Throughout the revision process, we have dedicated ourselves to preserving and, where possible, enhancing the quality of the manuscript and methods.

C: The methods are described in the main text, and the authors have made their code their analysis script available online. However, they should pin the versions of the different softwares in the conda environments of their snakemake pipeline to ensure the reproducibility.

R: We have included the version numbers of the tools in the Snakemake pipeline.

Response to Reviewer 3

C: The manuscript by Rozwalak et al presents the results of one of the first metagenomic studies of the ancient human gut virome. The authors used state of the art methods in order to extract, taxonomically classify and annotate viral contigs from the total community metagenome. The use of deamination data and the enrichment of sample for phages related to the mammalian (mostly human) gut phages, convincingly shows that the recovered genomes indeed belong to the ancient gut phages, and not to environmental phages contaminating the sample. High conservation level of Mushuvirus mushu infecting *F. prausnitzii* (one of the main species of the human gut commensals), is interesting and noteworthy. I believe this manuscript is of interest to the broad audience and is publishable in its present form.

R: We thank the Reviewer for appreciating our work, thoughtful review, and clear, detailed corrections.

C: I have only a couple of small comments:

Lines 207-208. Do authors assume here that the ancient *Mushuvirus mushu* is a direct predecessor of the modern virus? Given the diversity of gut phages, it is equally possible, if not more likely, that it is in fact a sister clade (different subspecies).

R: We agree with the Reviewer that a direct ancestral relationship is less likely than the scenario in which both strains originated from a shared ancestor. However, if analyzed genomes come from such sister clades, encountering two nearly identical strains is extremely unlikely because two independent lineages would typically accumulate more mutations than a single lineage. Even when considering a direct evolutionary connection, the probability of two *Mushuvirus mushu* strains having nearly 98% identity remains negligible. Thus, in our analyses, we assumed the simpler scenario to demonstrate that even the shortest evolutionary trajectory should lead to more divergent genomes. We have provided this explanation in the revised manuscript (lines: 221-224).

C: Lines 235-236: Family Rumicococcaceae is an older, deprecated name of the family Oscillospiraceae (<https://lpsn.dsmz.de/family/oscillospiraceae>)

R: Thank you for pointing out this inconsistency. We have updated the family name to *Oscillospiraceae* throughout the manuscript (line 257, Figure 4f).

Reviewer #1 (Remarks to the Author):

The authors have responded to all my queries and suggestions, adding in lots of detailed additional data, such as Table S6 and S7 which allows all ancient phages to be placed into context of modern phages, and Figure S1 which clearly shows Mushuvirus is special. Also, the additional detail and genome comparisons (Fig S4 S5) are fascinating, showing that gene order is largely preserved in the phages shown. The additional discussion of Mushuvirus replication is very welcome and wraps up the story well. For me, the manuscript is complete and the authors have done a great job on the revision, I recommend publication.

Reviewer #2 (Remarks to the Author):

The authors have addressed my previous comments in their revised version of the manuscript, and have updated the main text and supplementary material accordingly. No more comments from my side.